# Genomic Selection for Milk Production Traits in Xinjiang Brown Cattle

**DOI:** 10.3390/ani12020136

**Published:** 2022-01-07

**Authors:** Menghua Zhang, Hanpeng Luo, Lei Xu, Yuangang Shi, Jinghang Zhou, Dan Wang, Xiaoxue Zhang, Xixia Huang, Yachun Wang

**Affiliations:** 1College of Animal Science, Xinjiang Agricultural University, Urumqi 830052, China; zhangmenghua810@126.com (M.Z.); q609468041@163.com (L.X.); wangdan01100330@163.com (D.W.); zhangxiaoxue0726@163.com (X.Z.); 2Laboratory of Animal Genetics, Breeding and Reproduction, Ministry of Agriculture of China, National Engineering Laboratory of Animal Breeding, College of Animal Science and Technology, China Agricultural University, Beijing 100193, China; luohanpeng@cau.edu.cn; 3School of Agriculture, Ningxia University, Yinchuan 750021, China; shyga818@126.com (Y.S.); zhoujinghang@molbreeding.com (J.Z.)

**Keywords:** Xinjiang Brown cattle, milk, SCS, REML, Bayes, single-step GBLUP

## Abstract

**Simple Summary:**

Milk production is an important trait in the breeding and genetic improvement of Xinjiang Brown cattle. To obtain the best strategy for improving the reliability of the breeding value estimation for each trait, we used single-trait and multitrait models based on the A-array pedigree-based best linear unbiased prediction (PBLUP) and H-array single-step genomic best linear unbiased prediction (ssGBLUP) to perform the genetic evaluation of different strategies using the restricted maximum likelihood (REML) and Bayesian methods. Upon comparison, the ssGBLUP calculation results of the multitrait models obtained using the REML and Bayesian methods were better than those of other strategies. Considering the calculation time, the multitrait model REML method is recommended for ssGBLUP calculation to accurately predict the breeding value of young animals; thus, this strategy should be used for the early breeding selection of Xinjiang Brown cattle.

**Abstract:**

One-step genomic selection is a method for improving the reliability of the breeding value estimation. This study aimed to compare the reliability of pedigree-based best linear unbiased prediction (PBLUP) and single-step genomic best linear unbiased prediction (ssGBLUP), single-trait and multitrait models, and the restricted maximum likelihood (REML) and Bayesian methods. Data were collected from the production performance records of 2207 Xinjiang Brown cattle in Xinjiang from 1983 to 2018. A cross test was designed to calculate the genetic parameters and reliability of the breeding value of 305 daily milk yield (305 dMY), milk fat yield (MFY), milk protein yield (MPY), and somatic cell score (SCS) of Xinjiang Brown cattle. The heritability of 305 dMY, MFY, MPY, and SCS estimated using the REML and Bayesian multitrait models was approximately 0.39 (0.02), 0.40 (0.03), 0.49 (0.02), and 0.07 (0.02), respectively. The heritability and estimated breeding value (EBV) and the reliability of milk production traits of these cattle calculated based on PBLUP and ssGBLUP using the multitrait model REML and Bayesian methods were higher than those of the single-trait model REML method; the ssGBLUP method was significantly better than the PBLUP method. The reliability of the estimated breeding value can be improved from 0.9% to 3.6%, and the reliability of the genomic estimated breeding value (GEBV) for the genotyped population can reach 83%. Therefore, the genetic evaluation of the multitrait model is better than that of the single-trait model. Thus, genomic selection can be applied to small population varieties such as Xinjiang Brown cattle, in improving the reliability of the genomic estimated breeding value.

## 1. Introduction

Xinjiang Brown cattle was the first breed of cattle used for milk and meat purposes after the founding of the People’s Republic of China [1]. The breeding industry of Xinjiang Brown cattle accounts for a large proportion of the local economic development as well as farmers’ and herders’ income. In 2018, the population of Xinjiang Brown cattle reached 1.5 million [2]. However, compared with the Holstein cattle, genetic improvement technologies for Xinjiang Brown cattle are relatively behind. Xinjiang Brown cattle are a unique species in Xinjiang, where for a long time, the breeding value for the milk production traits of this cattle breed was estimated using pedigree-based best linear unbiased prediction (PBLUP) through the construction of the additive genetic relationship matrix (A matrix) [1]. However, with the reduction in sequencing cost, cattle breeding has entered the genome era [3].

The large-scale breeding of Xinjiang Brown cattle population is limited, and the number of the grazing or semihouse feeding and semi-grazing population is high. Although the production performance database of core breeding farms has been preliminarily established, the limitations of Xinjiang Brown cattle production performance data, coupled with the lack of pasture management and the presence of a significant amount of truncated data, have led to the genetic evaluation of Xinjiang Brown cattle in failing to achieve better reliability in terms of genomic estimated breeding value, thus influencing the accurate selection of excellent breeding animals. Compared with PBLUP, genome prediction can reduce the breeding cost and generation interval by at least 50% [4], leading to faster genetic progression. Genomic selection (GS) technology can also improve the reliability of genetic evaluation; therefore, it is imperative to apply it in genetically evaluating Xinjiang Brown cattle. A study on the GS of cattle mainly focuses on the reliability of models, methods, and genomic estimated breeding values [4]. On the basis of a previous calculation of genomic best linear unbiased prediction (GBLUP) using the genetic-pedigree joint relationship matrix (G matrix), single-step GBLUP (ssGBLUP) was proposed [5]. Furthermore, it is more comprehensive than GBLUP in genetically evaluating individuals [6]. ssGBLUP can simultaneously use the phenotype and pedigree information of genotype and non-genotype animals [7,8], and combine the A and G matrices to construct a genetic-pedigree relationship matrix (H matrix) [9,10,11] to perform a one-step genetic evaluation. For example, Li et al. [12] used a one-step method to estimate the milk production trait of Holstein cattle in China and increased the accuracy of estimation by 0.12 compared with the two-step GBLUP method. However, the study also found that adjusting the proportion of G and A matrices in the one-step method did not improve reliability [13,14]. Studies using different livestock species further confirmed that ssGBLUP was more reliable than BLUP and GBLUP in estimating GEBV [15,16,17,18].

The multitrait BLUP (MBLUP) has been studied for decades. It is a method for the genetic evaluation of individuals for two or multiple traits using information such as the phenotypic and genetic correlations of traits. An advantage of MBLUP [19] is the increased accuracy of genetic evaluation. The multitrait genetic evaluation of the REML and Bayesian methods is widely used in cattle breeding [20]. Studies have shown that the accuracy of the breeding value estimated by the multitrait model is twice that of PBLUP [21]. The accuracy of estimation based on the G-matrix multitrait model was also 3% higher than that of the single-trait model [22], where many studies have reported the advantages of multitrait genetic evaluation or GS [23,24,25,26].

Therefore, this study aimed to use the REML and Bayesian methods to estimate the genetic parameters of the milk production traits of Xinjiang Brown cattle using the A and H matrices, respectively, with a goal to ultimately obtain the variance components and genomic breeding values of each trait of the single- and multitrait models; this will improve the reliability of the estimated values and provide theoretical support for a more accurate estimation of the genetic parameters of the Xinjiang Brown cattle population.

## 2. Materials and Methods

### 2.1. Data Source and Processing

Data of Xinjiang Brown cattle were obtained from 7516 production performance measurement records and 16,795 pedigree records of the abovementioned four Xinjiang Brown cattle breeding pastures (The Xinjiang Tianshan Animal Husbandry Bio-Engineering Co.,Ltd, Xinjiang Uygur Autonomous Region local state-owned Urumqi cattle farm, Yili Xinjiang Brown cattle farm, Tacheng Agriculture and Animal Husbandry Technology Co. LTD, China) from 1983 to 2018 as well as DHI measurement records from 2010 to 2018. The pedigrees of Xinjiang Brown cattle breeding bulls, Xinjiang Brown cattle adult cows, and Swiss brown cattle breeding bulls were traced as well. This included 676 Xinjiang Brown cattle breeding bulls, among which 1 breeding bull had the most offsprings (619 offsprings), and 221 breeding bulls had only one offspring; 583 adult cows of Xinjiang Brown cattle had only one offspring, and 6199 adult cows of Xinjiang Brown cattle had ≥2 offsprings; the maximum number of offsprings was 12. The following milk production traits were obtained after sorting: 305 daily milk yield (305 dMY), milk fat yield (MFY), milk protein yield (MPY), and somatic cell score (SCS). SCC conversion to SCS is based on the formula determined by the American Dairy Cattle Improvement Program Board: $\mathrm{SCS}=log_{2}\left(\frac{SCC}{10^{5}}\right)+3$.

Effect Division: The field effect was divided into four levels, where the calving year was divided into seven levels according to phenotypic records: 1985–1995, 1996–2000, 2001–2005, 2006–2008, 2009–2011, 2012–2014, and 2015–2018; the calving season effect, according to the unique climatic conditions in Xinjiang, was divided into spring (April and May), summer (June, July, and August), autumn (September), and winter (January, February, March, October, November, and December) using the pentad mean temperature method; the birth order effect was divided into six levels: 1, 2, 3, 4, 5, and 6 (including >6 births).

### 2.2. Genotyping Data

The chip data included 403 Xinjiang Brown cattle cows, 71 Xinjiang Brown cattle bulls, and 11 Swiss brown cattle bulls, where a total of 139,376 single nucleotide polymorphism (SNP) loci were detected using the GeneSeek GGP Bovine 150 K chip (Illumina). The Beagle v.4.1 software was used for imputation, which mainly inferred the presence of haplotypes in the population based on the principle of linkage disequilibrium. Therefore, the quality of the chip data needed to be controlled to ensure the accuracy of imputation. Quality control standard: the SNP loci with an individual genotyping detection rate of <90%, individual genotype deletion rate of <10%, minor allele frequency of >0.01, and Hardy Weinberg equilibrium *p*-value of >1 × 10^−6^ were excluded. Finally, 118,021 SNP loci of Xinjiang Brown cattle were retained for subsequent analysis.

### 2.3. Statistical Analysis

#### 2.3.1. Estimation of Genetic Parameters of the Single-Trait Model

Different relationship matrices (*A* or *H* matrix) were constructed using single-trait models, PBLUP and ssGBLUP, and the genetic parameters of milk production traits of Xinjiang Brown cattle were estimated using the REML method.

Matrix expression formula of each trait model is shown below:Y=Xb+Za+e
where *Y* is the observed value vector of each trait (including 305 dMY, MFY, MPY, and SCS), *b* is the fixed effect vector (field, calving year, calving season, and birth order), *a* is the random additive genetic effect vector, *e* is the residual effect vector, *X* is the fixed effect coefficient matrix, and Z is the random additive genetic effect coefficient matrix.

For PBLUP [27] or ssGBLUP [28], assume $a\sim N\left(0,A\;\mathrm{or}\;H\sigma_{a}^{2}\right),\;e\sim N\left(0,I\sigma_{e}^{2}\right)$, where *A* is the additive genetic relationship matrix based on the pedigree, *H* is the genomic-pedigree joint relationship matrix, $\sigma_{a}^{2}$ represents additive genetic variance, *I* is the unit matrix, and $\sigma_{e}^{2}$ represents the variance of random error effect. With given *A* and *G* values, *H* is calculated using the following formula:$$H=\left[\begin{array}{cc} A_{11}-A_{12}A_{22}^{-1}A_{21}+A_{12}A_{22}^{-1}GA_{21} & A_{12}A_{22}^{-1}G\\ GA_{22}^{-1}A_{21} & G \end{array}\right]$$
where subscripts 1 and 2 of *A* represent the populations of non-genotyped and genotyped animals, respectively; *G* is the genomic relationship matrix, and its calculation formula is $G=\frac{MM^{\prime }}{2\sum_{k=1}^{m}p_{k}\left(1-p_{k}\right)}$; *M* is the incidence matrix of SNP effect, where elements $\left(0-2p_{j}\right)$, $\left(1-2p_{j}\right),$ and $\left(2-2p_{j}\right)$ represent 11 homozygotes, 12 or 21 heterozygotes, and 22 homozygotes, respectively;$\;p_{j}$ is the minimum allele frequency of the *j*th SNP; m is the number of markers; and $p_{k}$ is the allele frequency of the *K*th SNP. Therefore, the formula for *H*^−1^ is $H^{-1}=A^{-1}+\left[\begin{array}{cc} 0 & 0\\ 0 & G^{-1}+A_{22}^{-1} \end{array}\right]$, where $A^{-1}$ is the inverse matrix of all pedigree relationships; $G^{-1}$ is the inverse matrix of genomic relationship; and $A_{22}^{-1}$ is the inverse matrix of the pedigree relationship of genotyped individuals (the same letter in the multitrait model represents the same meaning).

#### 2.3.2. Estimation of Genetic Parameters of the Multitrait Model

The REML and Bayesian methods were used on the multitrait animal model to perform genetic parameter estimation based on PBLUP and ssGBLUP, where the matrix form of the multitrait model [29] is as follows:$$\left[\begin{array}{c} y_{1}\\ \begin{array}{c} y_{2}\\ y_{3} \end{array}\\ y_{4} \end{array}\right]=\left[\begin{array}{cccc} X_{1} & 0 & 0 & 0\\ 0 & X_{2} & 0 & 0\\ 0 & 0 & X_{3} & 0\\ 0 & 0 & 0 & X_{4} \end{array}\right]\left[\begin{array}{c} b_{1}\\ b_{2}\\ b_{3}\\ b_{4} \end{array}\right]+\left[\begin{array}{cccc} Z_{1} & 0 & 0 & 0\\ \vdots & Z_{2} & 0 & 0\\ 0 & 0 & Z_{3} & 0\\ 0 & 0 & 0 & Z_{4} \end{array}\right]\left[\begin{array}{c} a_{1}\\ a_{2}\\ a_{3}\\ a_{4} \end{array}\right]+\left[\begin{array}{c} e_{1}\\ e_{2}\\ e_{3}\\ e_{4} \end{array}\right]$$
where $y_{i}$ is the observed value vector of all individuals, $b_{i}$ is the fixed effect vector of the *i*th trait, $a_{i}$ is the additive genetic effect vector of the *i*th trait, $e_{i}$ is the residual random effect vector of the *i*th trait, and $X_{i}$ and $Z_{i}\;\mathrm{are}\;\mathrm{the}\;\mathrm{relationship}\;\mathrm{matrices}\;\mathrm{of}\;\beta$ and $a_{i}$, respectively. Assume $\left[\begin{array}{c} a_{1}\\ \ddots \\ a_{n} \end{array}\right]\sim N(0,A\;or\;H⊗\left[\begin{array}{ccc} \sigma_{a_{1}}^{2} & \cdots & \sigma_{a_{1}a_{n}}\\ \vdots & \ddots & \vdots \\ \mathrm{Symmetric} & \cdots & \sigma_{a_{n}}^{2} \end{array}\right]$, $\left[\begin{array}{c} e_{1}\\ \ddots \\ e_{n} \end{array}\right]\sim N(0,I⊗\left[\begin{array}{ccc} \sigma_{e_{1}}^{2} & \cdots & \sigma_{e_{1}e_{n}}\\ \vdots & \ddots & \vdots \\ \mathrm{Symmetric} & \cdots & \sigma_{e_{n}}^{2} \end{array}\right]$, where *A* is the additive genetic effect matrix; $\sigma_{a_{i}}^{2}$ and $\sigma_{e_{i}}^{2}$ are the additive genetic variance and random error effect variance of the *i*th trait, respectively; $\sigma_{a_{i}a_{j}}$ and $\sigma_{e_{i}e_{j}}$(*i* ≠ *j*) are the additive genetic covariance and random error effect covariance between the *i*th and *j*th traits, respectively. The formula for *H*^−1^ is $H^{-1}=A^{-1}+\left[\begin{array}{cc} 0 & 0\\ 0 & G_{W}^{-1}+A_{22}^{-1} \end{array}\right]$.

The REML method of the REMLF90 module of BLUPF90 software [30] was used to estimate the variance components. After the calculation results converged, AIREMLF90 was used to run for 0 times and the approximate standard errors of all calculation parameters were obtained based on the algorithm implemented by Meyer and Houle in the AIREMLF90 program. As an alternative of SE, calculate SD for function of (co)variances by repeated sampling of parameter estimates from their asymptotic multivariate normal distribution, following ideas presented by Meyer and Houle 2013 [31]. The Bayesian method was followed using the GIBBS1F90 module in the BLUPF90 software and the Bayesian Gibbs sampling method. In the Bayesian method, the total strand length of samples was 100,000 and the length of the preheating strand was 10,000, with a sparse interval of 50. The Geweke diagnostics method of POSTGIBBSF90 module of BLUPF90 software was used to check the convergence of the Gibbs chain. 

#### 2.3.3. Calculation of Heritability

The calculation formula of heritability is as follows:$$h^{2}=\frac{\sigma_{a}^{2}}{\sigma_{a}^{2}+\sigma_{pe}^{2}+\sigma_{e}^{2}}$$
where $h^{2}$ is heritability, $\sigma_{a}^{2}$ is the additive genetic variance, $\sigma_{pe}^{2}$ is the permanent environmental variance of individual, and $\sigma_{e}^{2}$ is the residual variance.

The formula for the standard error of heritability is shown below:$$SE^{2}(h^{2})=\left[\frac{\sigma_{a}^{2}}{\sigma_{p}^{2}}\right]\left[\frac{Var\left(\sigma_{a}^{2}\right)}{{\left(\sigma_{a}^{2}\right)}^{2}}+\frac{Var\left(\sigma_{p}^{2}\right)}{{\left(\sigma_{p}^{2}\right)}^{2}}-\frac{Cov\left(\sigma_{a}^{2},\sigma_{p}^{2}\right)}{\sigma_{a}^{2}\sigma_{p}^{2}}\right]$$
where $SE^{2}\left(h^{2}\right)$ is the standard error of heritability, $\sigma_{a}^{2}$ is the additive genetic variance, and $\sigma_{p}^{2}$ is the overall phenotypic variance, $\sigma_{p}^{2}=\sigma_{a}^{2}+\sigma_{pe}^{2}+\sigma_{e}^{2}$.

#### 2.3.4. Reliability of Breeding Value Estimation

The definition of reliability of GEBV was:$$R_{GEBV}^{2}=Cor^{2}\left(GEBV,a\right)=\frac{Cov^{2}\left(GEBV,a\right)}{V_{GEBV}V_{a}}$$

If GEBV is unbiased, a=GEBV+ε, $Cov\left(GEBV,\varepsilon \right)=0$, ε is prediction error.
$$Cov\left(GEBV,a\right)=Cov\left(GEBV,GEBV+\varepsilon \right)=V_{GEBV}\;R_{GEBV}^{2}=\frac{Cov^{2}\left(GEBV,a\right)}{V_{GEBV}V_{a}}=\frac{V_{GEBV}}{V_{a}}$$
where *a* and ε are independent, $R_{GEBV}^{2}$ is the reliability of genomic estimated breeding value, $Cor^{2}\left(GEBV,a\right)$ is the square of the correlation coefficient between genomic estimated breeding and actual values. The reliability of animal breeding values was calculated according to prediction error variance as
$$V_{a}=V_{GEBV}+PEV\;R_{GEBV}^{2}=\frac{V_{GEBV}}{V_{a}}=1-\frac{PEV}{V_{a}}$$
where *PEV* is the prediction error variance and $V_{a}$ is the direct additive genetic variance.

## 3. Results

### 3.1. Descriptive Statistical Analysis of Each Trait

Table 1 lists the sample number of observed values, minimum value, maximum value, mean value, standard deviation, and coefficient of variation of the milk production and reproductive traits of Xinjiang Brown cattle. Figure 1 shows that the data frequency distribution of the milk production traits of Xinjiang Brown cattle followed a normal distribution. The average milk production of Xinjiang Brown cattle in 305 days was 4216.49 kg, which was similar to that of Sanhe cattle and Chinese Simmental cattle but lower than that of dual-purpose cows such as European Fleckvieh cattle and Swiss brown cattle [32]. The mean MFY and MPY of the population were 168.53 kg and 143.79 kg, respectively, which were lower than those of Fleckvieh cattle and Swiss Brown cattle [33], and similar to those recently reported Italian Simmental cattle [34] SCS can be used as an indicator of breast health, where the lower the SCS value, the lower the risk for mastitis. These traits can reflect the production efficiency and health status of dairy cows and are also important goals for the breeding of Xinjiang Brown cattle.

### 3.2. Estimation of the Genetic Parameters of Milk Production Traits

Table 2 shows the estimation results of the genetic parameters for the milk production traits of Xinjiang Brown cattle obtained using the single-and multitrait models based on PBLUP and ssGBLUP using the REML or Bayesian methods. The heritability of milk production traits of Xinjiang Brown cattle that was calculated on the basis of PBLUP and ssGBLUP using the multitrait models REML and Bayesian methods was higher than those of each trait calculated using the single-trait model REML method; moreover, the standard error of heritability of the former model was lower than that of the latter. However, the heritability results of the milk production traits of Xinjiang Brown cattle calculated using the REML and Bayesian methods based on PBLUP and ssGBLUP of the multitrait model were similar; Gibbs parameter chains of additive genetic variance of milk production traits under different relationship matrices were obtained using the Bayesian method. Geweke diagnostic test results show that the convergence trend of iteration curve indicates that the parameter chain converges. The heritability of 305 dMY, MFY, and MPY estimated by the two methods in the multitrait model was approximately 0.39 (0.02), 0.40 (0.03), and 0.49 (0.02), respectively, which indicated high heritability traits (*h^2^* > 0.3). However, the heritability of SCS was approximately 0.07 (0.02), which suggested a low heritability trait (*h^2^* < 0.1).

### 3.3. Reliability of Breeding Value Estimation of Milk Production Traits

Table 3 compares the estimated breeding value (EBV), GEBV reliability, and increased reliability (Δreliability) of the overall and genotyped populations of Xinjiang Brown cattle that were calculated via different methods and models. Results showed that the reliability of GEBV calculated using ssGBLUP was higher than that of EBV calculated using PBLUP in the overall and genotyped populations regardless of whether it was based on the single- or multitrait model. Comparing with the multi- and single-trait models of the REML method, the reliability of EBV estimated using the multitrait model was higher than that estimated using the single-trait model; the reliability of GEBV estimated using the multitrait model was higher than that estimated using the single-trait model. However, comparing the REML and Bayesian methods of the multitrait model, the reliability of EBV and GEBV of SCS calculated using the Bayesian method was slightly higher than that calculated using the REML method in the overall and genotyped populations. In contrast, the EBV and GEBV reliability results of other traits were similar (Figure 2). In all results, the Δreliability of the genotyped population was higher than that of the overall population and the Δreliability of the Bayesian method was slightly higher than that of the REML method.

## 4. Discussion

### 4.1. Analysis of Genetic Parameters of Milk Production Traits

In this study and the one by Zhou Jinghang, the heritability of 305 dMY and SCS estimated using the REML method of the DMU software based on the pedigree data of Xinjiang Brown cattle was 0.40 (0.017) and 0.08 (0.009), respectively, which were both higher than the heritability results of PBLUP calculated using the REML and Bayesian methods in this study; the standard error of the former was lower. However, the heritability results were similar to those of ssGBLUP calculated using the REML and Bayesian methods in this study. The estimated heritability of MFY and MPY was 0.30 (0.013) and 0.20 (0.011), respectively, which were significantly lower than the results estimated using the REML and Bayesian methods for different genetic relationship matrices of the multitrait model in this study [35]. In addition, the heritability of somatic scoring was lower than that of the American and Italian brown cattle populations (0.12 [36] and 0.14 [37], respectively).

In this study, the heritability obtained using the REML and Bayesian methods in the multitrait model was higher than that in the single-trait model. Furthermore, in all model methods, except for the decrease in SCS, the heritability of other traits calculated based on ssGBLUP was slightly higher than that calculated based on PBLUP [29]; the standard errors of the heritability of the two were similar. The heritability in genetic parameter estimation based on the G matrix was lower than that based on the A matrix [38,39]. During ssGBLUP estimation, the slight increase in heritability was mainly reflected in the increase in additive genetic variance components [40,41]. The additive genetic variance difference between PBLUP and ssGBLUP estimations was primarily due to the construction of the A and H matrices being different, and the scale of diagonal elements was also different [42]. For example, the individual breeding value, $a_{i}=0.5\left(a_{s}+a_{d}\right)+m_{i}$, where $a_{s}$ is the breeding value of the father, $a_{d}$ is the breeding value of the mother, and $m_{i}$ is the Mendelian sampling deviation of individuals. For the A matrix, its diagonal is the expected value of the genetic relationship coefficient between individuals. In contrast, the diagonal in the G matrix is the real genetic relationship coefficient between individuals considering the Mendelian sampling deviation [43,44]. For example, when using paternal or maternal phenotypic information to estimate the breeding value, if the genetic relationship of the A matrix is 0.5, the genetic relationship of the G matrix will be ≥0.5 or ≤0.5; the H matrix combines the A and G matrices. Several studies have shown that GBLUP and ssGBLUP are better than PBLUP. The main reason is that the G matrix shows a more realistic genetic relationship between individuals than the A matrix [44,45,46]. Therefore, the parameter estimation of the A and H matrices may have deviations [47]. In addition, many studies conducted in China and overseas have confirmed that the REML method is relatively ideal for estimating genetic parameters of livestock breeding [48]. As the Bayesian method can consider prior information of unknown parameters and provide accurate posterior distribution for a limited sample size [49], it is favorable for the genetic evaluation of small breed populations when there is a large amount of historical data available [50]; however, its calculation time is relatively long.

### 4.2. Predictive Analysis of EBV and GEBV Reliability

When using the REML method of the single-trait model, only the reliability of the breeding value of 305 dMY was >0.4. The reliability of the breeding value of milk production traits estimated based on PBLUP and ssGBLUP of the REML and Bayesian methods in the multitrait model ranged from low to moderate. In general, the increased ratio of the reliability of genotyped population was 0.9%–3.6% higher than that of the overall population, and the GEBV reliability of the genotyped population was up to 83%. Compared with PBLUP, ssGBLUP method with chip information added significantly and simultaneously improved the reliability of breeding value estimation in both REML and Bayesian methods (Figure 2), which indicated that GS using ssGBLUP is feasible for the genetic evaluation of Xinjiang Brown cattle. Although the Bayesian method of the multitrait model is slightly better than the REML method, using the REML method to calculate ssGBLUP can save more calculation time.

Relying on the linkage disequilibrium between SNP markers and quantitative trait locus of target traits, genomic prediction promotes the association between SNP markers and individual phenotypic values. Thus, the accurate estimation of SNP’s genetic effect based on the phenotypic value has become a key factor in genomic prediction; the accuracy of genomic prediction [51] can be improved by increasing the number and accuracy of phenotypic values (by expanding the reference population) [52]. Genotyping candidate animals with different SNP loci and chip densities can also influence the accuracy of GS [53]. In the ssGBLUP method, the genomic breeding value estimation accuracy is higher for traits with a larger number of animal phenotypes and genotypes and those with high heritability [54].

With increased reference population size and improved pedigree data integrity, the accuracy of EBV or genomic breeding estimation increases [55]. However, there are relatively few genotyped populations of Xinjiang Brown cattle (genotyped individuals account for only 2.89%), which is the key factor restricting the reliability of ssGBLUP estimation of breeding value. Furthermore, some studies have reported that the reliability of low heritability traits is greatly improved when GBLUP is used [56]. In studies using different livestock [15,17,57,58,59], the same conclusion that ssGBLUP can more accurately predict the genetic value of animals than the classical PBLUP method was drawn. In addition to increasing the number of reference groups of this variety, some studies have shown that the combined reference population significantly improves the reliability of GEBV for the populations of four countries. Compared with the use of the domestic population alone as a reference, the reliability of GEBV is increased by an average of 10% when different reference population sizes are used for different countries and traits. In addition, expanding the reference population size can significantly improve the reliability of GEBV [50] by 2–19% [52,60]. Therefore, in the future, the cross variety GS of Xinjiang Brown cattle can also be used as a strategy to expand the reference population and improve the reliability of the breeding value estimation.

## 5. Conclusions

It is feasible to use ssGBLUP to perform the genomic evaluation of the milk production traits of Xinjiang Brown cattle. However, the single-trait model does not consider the covariance between traits, and thus has a large error. The use of the multitrait model, regardless of the use of PBLUP or ssGBLUP, can greatly improve the reliability of the breeding value estimation. Moreover, the estimated genetic parameters provide a basis for calculating a more accurate breeding value of Xinjiang Brown cattle. However, if it is needed to obtain a higher level of breeding value reliability, the number of genotypes needs to be simultaneously expanded when using high-density chips (150 k). Nevertheless, applying the ssGBLUP method to the genetic evaluation of Xinjiang Brown cattle necessitates further study and analysis. The next challenge is to build a joint reference group through cross variety GS to expand the number of genotyped animals with the objective to improve the reliability of the breeding value estimation of various traits of Xinjiang Brown cattle.

## Figures and Tables

**Figure 1 animals-12-00136-f001:**
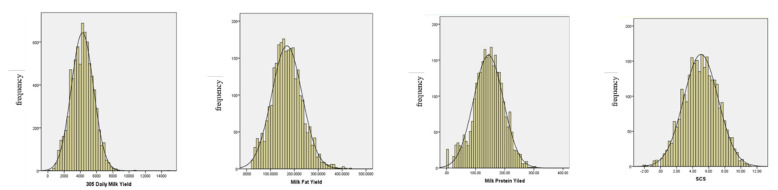
Frequency distribution of phenotypic data of Xinjiang Brown Cattle.

**Figure 2 animals-12-00136-f002:**
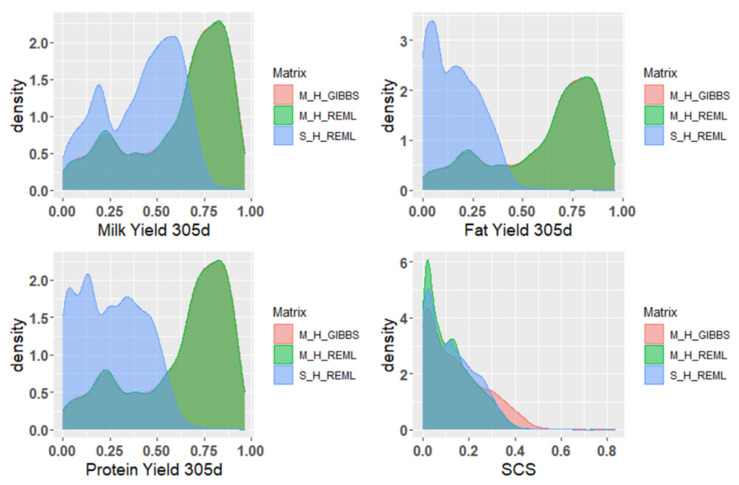
Density map of the reliability of GEBV using different methods of the single-trait and multitrait models. M_GIBBS_H is the reliability of GEBV using the Bayesian method of the multitrait model, M_REML_H is the reliability of GEBV using the REML method of the multitrait model, and S_REML_H is the reliability of GEBV using the REML method of the single-trait model.

**Table 1 animals-12-00136-t001:** Description of milk trait in Xinjiang Brown Cattle.

Trait ^1^	Number	Minimum	Maximum	Average	SD	CV
305 dMY/kg	7515	814	8444	4126.49	1405.71	34.07
MFY/kg	2655	21.6	431.55	168.53	68.29	40.52
MPY/kg	2655	20.3	302.72	143.71	51.42	35.78
SCS	2655	−2.05	10.95	4.98	2.16	43.37

^1^ 305 dMY: 305 daily milk yield; MFY: milk fat yield; MPY: milk protein yield; SCS: somatic cell score.

**Table 2 animals-12-00136-t002:** Variance components and heritability of milk traits obtained using the pedigree relationship matrix (pedigree-based best linear unbiased prediction (PBLUP)) and combined genomic-pedigree matrix (single-step genomic best linear unbiased prediction (ssGBLUP)) (SE of variance components and heritability reported in parentheses).

		PBLUP			ssGBLUP		
	Trait ^1^	$\sigma_{a}^{2}(\mathrm{SE})$	$\sigma_{e}^{2}(\mathrm{SE})$	$h^{2}(\mathrm{SE})$	$\sigma_{a}^{2}(\mathrm{SE})$	$\sigma_{e}^{2}(\mathrm{SE})$	$h^{2}(\mathrm{SE})$
REML (single-trait model)	305 dMY	275,620 (21,393)	922,930 (18,862)	0.238 (0.016)	276,960 (21,545)	924,850 (18,869)	0.239 (0.016)
	MFY	198.390 (67.272)	2849.800 (95.395)	0.065 (0.022)	197.690 (63.325)	2843.100 (93.252)	0.065 (0.021)
	MPY	233.960 (43.413)	1427.100 (47.843)	0.141 (0.025)	230.860 (43.347)	1429 (47.566)	0.139 (0.025)
	SCS	0.177 (0.076)	4.0239 (0. 127)	0.042 (0.018)	0.15410 (0.073)	4.047 (0.127)	0.037 (0.017)
REML (Multiple-trait model)	305 dMY	499,900 (30,746)	803,200 (15,783)	0.384 (0.016)	507,300 (31,206)	804,400 (15,803)	0.387 (0.016)
	MFY	1341 (119.960)	2138 (67.600)	0.386 (0.024)	1368 (121.450)	2146 (67.735)	0.389 (0.024)
	MPY	937.500 (74.342)	1026 (1.370)	0.478 (0.023)	961.400 (75.724)	1028 (31.676)	0.483 (0.027)
	SCS	0.189 (0.077)	4.015 (0.130)	0.045 (0.018)	0.164 (0.07362)	4.040 (0.127)	0.039 (0.017)
Bayes (Multiple-trait model)	305 dMY	506,620 (26,877)	803,370 (15,108)	0.387 (0.014)	503,920 (26,790)	805,970 (15,165)	0.385 (0.014)
	MFY	1368.800 (101.480)	2142.800 (63.356)	0.389 (0.020)	1426.400 (100.910)	2148.500 (63.497)	0.399 (0.019)
	MPY	932.260 (53.003)	1031.300 (30.717)	0.475 (0.017)	987.080 (59.840)	1032.300 (30.720)	0.488 (0.018)
	SCS	0.290 (0.080)	3.978 (0.129)	0.068 (0.018)	0.273 (0.078)	3.997 (0.129)	0.065 (0.018)

^1^ 305 dMY: 305 daily milk yield; MFY: milk fat yield; MPY: milk protein yield; SCS: somatic cell score. $\sigma_{a}^{2}$: additive genetic variance; $\sigma_{e}^{2}$ residual variance; *h*^2^: heritability; SE: standard error.

**Table 3 animals-12-00136-t003:** Comparison of estimated breeding value (EBV) and genomic estimated breeding value (GEBV) reliability, Spearman’s rho (PBLUP and ssGBLUP) and increases in reliability (Δrel) obtained for the whole population and the genotyped subpopulation by different methods in Xinjiang Brown Cattle.

		Whole Population				Genotyped Subpopulation			
	Traits ^1^	PBLUP	ssGBLUP	Δrel (%)	Correlation	PBLUP	ssGBLUP	Δrel (%)	Correlation
REML (single-trait model)	305 dMY	0.404 (0.206)	0.414 (0.201)	1	0.98 **	0.491 (0.123)	0.526 (0.108)	3.5	0.89 **
	MFY	0.148 (0.117)	0.166 (0.126)	1.8	0.89 **	0.213 (0.115)	0.237 (0.115)	2.5	0.73 **
	MPY	0.242 (0.172)	0.258 (0.173)	1.6	0.94 **	0.341 (0.154)	0.377 (0.150)	3.6	0.81 **
	SCS	0.115 (0.097)	0.125 (0.106)	1	0.87 **	0.161 (0.093)	0.172 (0.095)	1.1	0.76 **
REML (Multiple-trait model)	305 dMY	0.612 (0.265)	0.620 (0.257)	0.9	0.99 **	0.811 (0.092)	0.825 (0.085)	1.4	0.97 **
	MFY	0.610 (0.265)	0.619 (0.257)	1	0.99 **	0.810 (0.092)	0.824 (0.085)	1.4	0.97 **
	MPY	0.611 (0.265)	0.619 (0.257)	0.9	0.99 **	0.810 (0.092)	0.825 (0.085)	1.5	0.97 **
	SCS	0.109 (0.099)	0.119 (0.103)	1	0.90 **	0.190 (0.095)	0.199 (0.097)	1	0.81 **
Bayes (Multiple-trait model)	305 dMY	0.614 (0.265)	0.621 (0.258)	0.8	0.99 **	0.813 (0.090)	0.828 (0.083)	1.5	0.97 **
	MFY	0.610 (0.264)	0.619 (0.257)	1	0.99 **	0.809 (0.090)	0.825 (0.083)	1.6	0.97 **
	MPY	0.613 (0.265)	0.621 (0.258)	0.9	0.99 **	0.812 (0.090)	0.827 (0.083)	1.5	0.97 **
	SCS	0.133 (0.122)	0.146 (0.128)	1.3	0.91 **	0.235 (0.118)	0.260 (0.119)	2.5	0.79 **

^1^ 305 dMY: 305 daily milk yield; MFY: milk fat yield; MPY: milk protein yield; SCS: somatic cell score; SE: standard error; ** indicates correlation being significantly different from 0 (*p <* 0.01).

## Data Availability

The study did not report any data.

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
