# Peer review of "Genomic Selection for Milk Production Traits in Xinjiang Brown Cattle"

_animals, 2022, doi:10.3390/ani12020136_

Round 1

Reviewer 1 Report

The manuscript reports the application of single-trait and multi-trait models for accurate prediction of the heritability and breeding value of four traits important in milk production in Xinjiang brown cattle. The authors carried out significant experimental and computational work. Their data base with the production performance measurements and the pedigree records is impressive as well as their genotyping data. The analysis of the descriptive statistics for each trait and the estimation of the genetic parameters of the traits associated with milk production are performed correctly. Generally, it is an interesting manuscript representing some interest in evaluating the milk production traits.

The manuscript requires minor revision.

1) The manuscript requires moderate editing of English language and style. There are many typos across the manuscript. The authors need to review the entire manuscript text and the legends in the figures and tables.

2) Simple Summary: Review the marks of punctuation.

3) Page 2, lines 52 and 53: Replace “species” by “breed”.

4) Figure 1, page 6. I like this illustration. However, please, make the text in this Figure readable and the y-axis legend in English.

5) Page 9, Line 320: Replace Figure 1 by Figure 2.

6) References. The authors need to follow the same style in the references. Compare inconsistencies in using capital and small letters. Please, review all your references.

7) Are the references 1, 19, 50, 55 books, research reports, or something else? How could readers find their original texts?

Author Response

Response to Reviewer 1 Comments

Thank you for your hard review and good suggestions. I value your opinions and suggestions very much. The following modifications have been made to the article according to your comments and suggestions:

Point 1: The manuscript requires moderate editing of English language and style. There are many typos across the manuscript. The authors need to review the entire manuscript text and the legends in the figures and tables. 

Response 1: Revise mistakes and legends in the manuscript.

Point 2: Simple Summary: Review the marks of punctuation. 

Response 2: Revise punctuation marks in the manuscript.

Point 3: Page 2, lines 52 and 53: Replace “species” by “breed”.

Response 3: Change “species” to “breed”.

Point 4: Figure 1, page 6. I like this illustration. However, please, make the text in this Figure readable and the y-axis legend in English.

Response 4: Figure 1, page 6. make the y-axis legend in English..

Point 5: Page 9, Line 320: Replace Figure 1 by Figure 2.

Response 5: Page 9, Line 320: Replace Figure 1 by Figure 2.

Point 6: References. The authors need to follow the same style in the references. Compare inconsistencies in using capital and small letters. Please, review all your references.

Response 6: Authors of references use consistent capital and small letters.

Point 7: Are the references 1, 19, 50, 55 books, research reports, or something else? How could readers find their original texts?

Response 7: the references 1, 50, 55 are Master's and Doctor's thesis, the references 19 is a book. They can be found in the following ways:

the references 1 (This is a master's thesis of my school, which was graduated in May 2020. It has not been officially uploaded to CNKI.)

the references 19:

https://baike.so.com/doc/9948278-10295772.html

the references 50:

http://202-112-175-70.vpn.cau.edu.cn:8118/Thesis65/SingleSearch/index?dbID=5&dbCode=Empty5

the references 55:

https://kns.cnki.net/kcms/detail/detail.aspx?dbcode=CDFD&dbname=CDFDLAST2019&filename=1018275005.nh&uniplatform=NZKPT&v=2eze3uorIdptmyiYgBjcOGsOGwj1zP_E9XZYGLUunRZYR4fU6NrlQDDD2YsZYQ1f

Have a lovely day!

Kind regards,

Menghua Zhang

Reviewer 2 Report

Experiments have been conducted rigorously. The manuscript is presented in an intelligible fashion, technically sound, and the data support the conclusions.

The authors used in their work different methods to estimate the genetic parameters of the milk production traits of Xinjiang brown cattle in order to obtain the variance components and genomic breeding values of each trait of the single and multitrait models. This is a laudable work that will improve the reliability of the estimated values and provide theoretical support for a more accurate estimation of the genetic parameters of Xinjiang brown cattle population, that could lead to a very nice publication.

The most accurate estimation of the genetic parameters of the cattle milk production traits plays a important role in breeding and genetic evaluation. Therefore, the work is of importance at regional level, and the information could prove useful to researchers and policy-makers in the field.

The introduction provides all important information of the general problem that the article addresses including highlights from previous research and the study purpose. The methodology section is sufficiently detailed and the study design and statistical analysis methods are appropriate, given the study purpose. The results section is logical presented following the analysis plan described in the methods section. All conclusions are supported by the results and the references are appropriate. In conclusion, experiments have been conducted rigorously. The manuscript is presented in an intelligible fashion, technically sound, and the data support the conclusions. 

I recommend the manuscript to be considered for publication.

Author Response

Thank you very much for your hard review, and thank you even more for your affirmation of the article. I value your opinions and suggestions very much. Wish you a pleasant work.

Have a lovely day!

Kind regards,

Menghua Zhang

Reviewer 3 Report

Dear authors,

the aim of the manuscript is of interest and the paper reads pretty well.

I have just one major concern about the validation of the models, that is not very clear (see the line-by-line comment below).

I also highlighted few minor comments (please see below).

Best regards

Line-by-line comments

Line 46: “are the first breed” should be “is the first breed”

Line 76: “Li used a one-step” is a reference? If so please consider writing “Li et al. [12] used a one-step”

Lines 194-204: How did you perform the validation? Which animals did you consider for the validation? Only females, right? Or you computed these correlations for all animals? If so explain and justify. Some examples that you can use as references for the female validation: https://doi.org/10.3168/jds.2020-19789, https://doi.org/10.2527/jas.2014-8836, https://doi.org/10.3168/jds.2020-19789, https://doi.org/10.3168/jds.2019-17963

Lines 213-215: Production levels are similar to those recently reported for Italian Simmental cattle breed (please see https://doi.org/10.3168/jds.2019-17421)

Line 219, Table 1: How did you compute SCS values? Are those average per lactation? What do you mean with a SCS value of “-2.05” ?

Line 232: “Geweke diagnostic test was performed using the POSTGIBBSF90 software” this is more materials and methods than results

Line 349: Why do you have a summary here? Remember that conclusions section is mandatory.

Lines 352-353: “Furthermore, using the ssGBLUP method with genomic information cannot improve the reliability of the breeding value estimation of milk production traits.” This sentence makes no sense; ssGBLUP is of course with genomic information, and before you concluded that reliabilities under ssGBLUP were higher. Please check this sentence.

Author Response

Thank you for your hard review and good suggestions. I value your opinions and suggestions very much. The following modifications have been made to the article according to your comments and suggestions:

Point 1: Line 46: “are the first breed” should be “is the first breed”. 

Response 1: “are the first breed” has been changed to “is the first breed”.

Point 2: Line 76: “Li used a one-step” is a reference? If so please consider writing “Li et al. [12] used a one-step”.

Response 2: “Li used a one-step” has been changed to “Li et al. [12] used a one-step”.

Point 3: Lines 194-204: How did you perform the validation? Which animals did you consider for the validation? Only females, right? Or you computed these correlations for all animals? If so explain and justify. Some examples that you can use as references for the female validation: https://doi.org/10.3168/jds.2020-19789, https://doi.org/10.2527/jas.2014-8836, https://doi.org/10.3168/jds.2020-19789, https://doi.org/10.3168/jds.2019-17963

Response 3: Lines 194-204: I computed these correlations for all animals, The correlation between the estimated genomic breeding value and phenotype was calculated to determine the accuracy of the genomic prediction equation. The square of the accuracy was the reliability. The reliability of animal breeding values was calculated as:

where a and e are independent,  is the reliability of breeding value estimation,  is the correlation between genomic breeding value and phenotypic value (corrected phenotypic value, the value after subtracting the fixed effect portion of the phenotypic value.), and  is the square of the correlation coefficient between estimated genomic breeding and actual values.

Point 4: Lines 213-215: Production levels are similar to those recently reported for Italian Simmental cattle breed (please see https://doi.org/10.3168/jds.2019-17421)

Response 4: added “ similar to those recently reported Italian Simmental cattle.” References were added. Thank you for the references.

Point 5: Line 219, Table 1: How did you compute SCS values? Are those average per lactation? What do you mean with a SCS value of “-2.05” ?

Response 5: SCC conversion to SCS is based on the formula determined by the American Dairy Cattle Improvement Program Board:  . The somatic cell score was -2.05 because the somatic cell number was 3.02Thousands/mL, and the formula was converted to -2.05.

Point 6: Line 232: “Geweke diagnostic test was performed using the POSTGIBBSF90 software” this is more materials and methods than results.

Response 6: “Geweke diagnostic test was performed using the POSTGIBBSF90 software”, this part goes into the materials method.

Point 7: Line 349: Why do you have a summary here? Remember that conclusions section is mandatory.

Response 7: The original summary was the conclusion part, which was translated with the wrong words.

Point 8: Lines 352-353: “Furthermore, using the ssGBLUP method with genomic information cannot improve the reliability of the breeding value estimation of milk production traits.” This sentence makes no sense; ssGBLUP is of course with genomic information, and before you concluded that reliabilities under ssGBLUP were higher. Please check this sentence.

Response 8: Delete the “Furthermore, using the ssGBLUP method with genomic information cannot improve the reliability of the breeding value estimation of milk production traits.”

Have a lovely day!

Kind regards,

Menghua Zhang

Round 2

Reviewer 3 Report

Dear Authors,

I'm satisfied by almost all your answers and changes. However, I still have a main concern about the validation. How did you compute the correlation between (G)EBV and phenotypes for all the animals? Because bulls do not have phenotypes you can use for the correlation.

Please fix the style of the reference you added.

Author Response

Response to Reviewer 3 Comments

Thank you for your hard review and good suggestions. I value your opinions and suggestions very much. The following modifications have been made to the article according to your comments and suggestions:

Point 1: I'm satisfied by almost all your answers and changes. However, I still have a main concern about the validation. How did you compute the correlation between (G)EBV and phenotypes for all the animals? Because bulls do not have phenotypes you can use for the correlation.Please fix the style of the reference you added.

Response 1: I apologize for the wrong for the misunderstanding, and I am so lucky to meet you such a professional and wise reviewer as you, thank you very much. I have checked all the formulas and contents of the manuscript, and updated the calculation formula for practical application.

Thank you again for your valuable advice, which helped me correct this mistake, and I apologize again for my carelessness.

Have a lovely day!

Kind regards,

Menghua Zhang
